# Moral Distress Events and Emotional Trajectories in Nursing Narratives during the COVID-19 Pandemic

**DOI:** 10.3390/ijerph19148349

**Published:** 2022-07-08

**Authors:** Daniela Lemmo, Roberta Vitale, Carmela Girardi, Roberta Salsano, Ersilia Auriemma

**Affiliations:** Department of Humanities, University of Naples Federico II, 80100 Naples, Italy; roberta.vitale9@studenti.unina.it (R.V.); carm.girardi@studenti.unina.it (C.G.); r.salsano@studenti.unina.it (R.S.); ersiliauriemma@virgilio.it (E.A.)

**Keywords:** moral distress, nursing, COVID-19 pandemic, narratives, autobiographical memories, emotions, clinical psychological intervention

## Abstract

The COVID-19 pandemic produced several ethical challenges for nurses, impacting their mental health and moral distress. In the moral distress model the categories of events related to moral distress are: constraint, dilemma, uncertainty, conflict, and tension, each one related to different emotions. This study explored moral events’ memories and emotions in narratives of a sample of 43 Italian nurses who worked during the COVID-19 pandemic. We constructed an ad-hoc narrative interview asking nurses to narrate the memory, and the associated emotion, of an event in which they felt they could not do the right thing for the patient. We conducted a theory-driven analysis, using the categories proposed by the literature, identifying the main emotion for each category. Results show that 36 memories of events are representative of moral distress; among these, 7 are representative of none of the categories considered, and we categorized them as moral compromise. The main emotional trajectories are powerlessness, worthlessness, anger, sadness, guilt, and helplessness. From a clinical psychological point of view, our findings highlight the narration of the memories of moral events as a tool to use in the ethical sense-making of critical experiences, in order to promote well-being and moral resilience among nurses in emergency situations.

## 1. Introduction

The COVID-19 pandemic was an unprecedented global health emergency, which required healthcare contexts to reorganize departments and healthcare providers; coping with adaptation processes to new forms of stressors [1,2] resulted in high levels of frustration, discrimination, and isolation [3,4] that negatively impacted the psychological well-being of the many nurses involved in its management [5,6].

The virus, totally unknown before the pandemic, generated a very high number of infections and critical conditions, and nurses found themselves dealing with a series of sudden changes. The time available to act was limited, the organizational resources available were insufficient, and the knowledge of the procedures to be implemented was still not widespread [7,8,9].

Healthcare organizations always represent an environment of psycho-physical risk for workers who are exposed on a daily basis to the suffering of their patients and to the potential failure of care [10]; in this respect, the spread of the COVID-19 epidemic certainly added a series of new risk factors to the numerous responsibilities of nurses, transforming the health emergency into a full-fledged mental health emergency. Interminable shifts difficult to reconcile with natural circadian rhythms [11], concerns associated with the effectiveness of personal protective equipment while exposed to the virus during working hours, and widespread fear of being infected and, thus, infecting their own families [12,13] represent only some of the risk conditions for the development of psychological disorders, or cases of mental, physical, and emotional distress. Some studies show positive correlations between anxiety levels, sleep quality, and problem-solving skills [14,15], and high levels of anxiety, depression [16], post-traumatic stress, and emotional exhaustion [17] in nurses involved on the front line of the COVID-19 emergency.

Several challenges brought to nurses by the pandemic are of an ethical and moral nature [18,19]. Maintaining high standards of moral behavior, which is usually the basis of the nursing profession, proved very difficult because of insecurity, high-risk situations, and the numerous anti-COVID-19 regulations to be respected. The lack of personal protective equipment [20], the lack of beds in intensive care departments, the lack of staff [19], the impossibility for relatives to visit patients, and the obstacles in sharing the work, all required a balance between professional duties, skills, and urgent ethical choices to be put in to practice.

### Moral Distress Memories and Emotions in Ethical Sense-Making

When nurses feel they are unable to act ethically in morally demanding situations, they may experience emotions of distress, which are identified with the construct of “moral distress” [21,22,23,24], particularly studied during the pandemic [25,26,27,28,29].

In this paper, we refer to the broader conception of moral distress proposed by authors of moral distress model [18,19,20,21,22,23,24,25,26,27,28,29,30], considering five categories of moral events that highlight various experiences of distress related to the ethics of nurses; these are events that can generate different negative emotions grouped under the umbrella term of distress.

The categories of moral events and the emotions associated with them are:*Moral constraint*: The moral agents think or feel that they know what is the most ethical action to take, but obstacles, real or perceived, internal or external, prevent them from taking it. The main emotions associated with this event are anger, frustration, powerlessness, and guilt;*Moral tension*: The moral agents feel or think they know what is the right thing to do, but do not share their point of view with others who think otherwise, and, therefore, do not engage in conflict. Therefore, they do not transform their moral principle into action, nor do they share it with colleagues. Anger, frustration, powerlessness, and guilt are frequently experienced emotions. Moral tension events can be precursors of moral conflict events;*Moral conflict*: The moral agents feel or think they know the right thing to do, and engage in conflict with people who have a different view on the decision to make. Moral conflict concerns an external and relational dimension and anger, frustration, powerlessness, and sadness are frequently experienced emotions;*Moral dilemma*: The moral agents feel or believe that there are two or more morally adequate actions, which are mutually exclusive. This means that nurses can choose to act in accordance with a moral principle, and, at the same time, feel that they have not respected another. Often a moral residue is present, along with emotions of guilt, strain, frustration, and sadness;*Moral uncertainty*: The moral agents cannot choose which moral principle to fulfil. Strain, frustration, and guilt are often present.

The resulting emotional conditions represent a source of discomfort for nurses, with the risk of a series of consequences on a physical, spiritual, behavioral, and emotional level [31,32,33,34,35,36]. Moral distress is associated with lower self-esteem, a decrease in work satisfaction and commitment [37,38,39], and a decrease in satisfaction with the quality of assistance provided to the patient [40], until the intention to resign or effectively leave work are reached [38,39,40,41].

Therefore, we affirm that, concerning moral issues, it is not possible to neglect the emotional dimension of the individual. The latter, in addition to being an integral part of moral reasoning, actively accompanies the processes of signification of experiences, through which the building blocks for the construction of a coherent vision of the self and the external world are added.

From a socio-constructivist perspective, we wonder how the relationship between moral events and associated emotions informs the creation of the ethical sense, in particular through the narration of the memory of the event itself. When we talk about creating ethical sense, we are referring to mental processes concerned with making sense of, and building meaning in relation to, moral issues [42].

Narration has always been considered an important means through which people can continuously reconstruct the meaning of autobiographical events [43,44]. In fact, by telling and, therefore, producing a story, a person refines some details of the event to the advantage of others: some aspects become more significant than others, and the need for coherence and continuity is satisfied [45], reconstructing the missing parts and repairing any fragmentations in the story [46]. In autobiographical memories, a crucial role must be attributed to emotions, and, therefore, to the emotional content of memories, of which one can acquire awareness through the narrative form [47]. An important tradition of research on memory and narration suggests that memories, after being narrated, do not maintain the same characteristics as when they are simply retrieved without a linguistic aid [45,48]; thus, narrated emotional memories are significantly more complex than the untold ones [49]. Not only does autobiographical narration allow for a richer and more complex emotional expression, but this expression seems to have beneficial effects on physical and emotional health [50,51,52].

Within a qualitative research design, in this study we explore the experience of moral distress in a sample of Italian nurses who worked with COVID-19 patients during the pandemic. In particular, we intend to analyze the narratives of memories of morally stressful events, starting from the categories of moral distress events [18,19,20,21,22,23,24,25,26,27,28,29,30], and then highlight the emotion most frequently described by nurses in relation to the memories of such events.

This study gives the opportunity to discuss some preliminary clinical reflections on the construction of personalized support interventions for nurses regarding moral distress events during the COVID-19 pandemic.

## 2. Materials and Methods

### 2.1. Tools and Participants

The research was conducted online due to the COVID-19 emergency state.

The nurses were identified through Facebook groups and contacted by psychologist researchers who, after presenting the objectives and methods of the research project, asked if they were available to participate by means of a telephone interview talking about their work experience during the COVID-19 pandemic. All interviews were conducted in March and April 2021.

The inclusion criteria for the enrolment in the research required that the participants had worked or still worked within a COVID-19 ward of an Italian hospital, or that they had served in the Italian Emergency Health system 118 during the pandemic.

Each nurse provided a telephone contact for conducting the narrative interview, which was audio recorded, and subsequently transcribed verbatim. The nurses’ contribution was voluntary; each participant signed an informed consent for their enrolment in the study, and a document for the protection of their privacy in accordance with the GDPR EU 2016/679, D.L. 101/2018. The study was conducted in accordance with the Declaration of Helsinki, and approved by the Ethics Committee of the Psychological Research of the Department of Humanities of the University of Naples Federico II (Prot.n. 21/2021).

In order to understand the meaning attributed by nurses to moral events experienced during the pandemic, and the associated emotions, in this study we adopted a qualitative research design, and constructed an ad-hoc narrative interview. We created the narrative interview with some areas to explore, and tried to encourage a gradual immersion of the nurses in the story of their professional experience. The interview aimed to bring out the memory of a specific morally critical event.

The first area of the interview concerned the story of nursing in the context of the COVID-19 pandemic (*Can you tell me how long have you been a nurse and how long have you been working with COVID patients? How has your work experience changed during the pandemic?*).

The second area concerned deep memories of moral events, and it aimed at exploring the autobiographical memory of a moral event, linked to the relationship with COVID-19 patients, which nurses consider significant for them and for their professional role (*Can you tell me an episode during which you felt that you could not do the right thing for the patient?*).

The third area concerned the emotional component of memory, and it aimed at exploring the main emotion described in relation to the memory of a morally critical event (*Can you tell me the main emotion that comes to your mind thinking about the episode you told me?*).

The entire interview lasted about 20 min.

### 2.2. Methods of Data Analysis

The coding of the narrative corpus was conducted through a theory-driven approach [53], starting from the 5 categories of moral events proposed by moral distress model [18,19,20,21,22,23,24,25,26,27,28,29,30]. The analysis was conducted by 4 independent judges, and proceeded through several steps:Shared reading of all the narratives, in order to identify which of the events narrated were representative of a condition of moral distress, in accordance with the characteristics described by the literature;Coding of representative narratives of events of moral distress within one of the categories of moral events [18,19,20,21,22,23,24,25,26,27,28,29,30];For each category of moral distress events, the way in which the nurses’ memories were narrated was identified, highlighting the main morally demanding situations around which the ethical sense-making is articulated;Identification of the main emotions mentioned by the nurses in relation to the event of moral distress they narrated.

The 4 independent judges came to a 99% agreement on the coding of all interviews.

## 3. Results

We recruited 42 nurses (16 men, 17 women) who worked, or were still working, with COVID-19 patients in various Italian regions: 44% of them work in southern Italy; 51% in regions of northern Italy, and 5% in regions of central Italy. A total of 11% of the nurses work at the emergency–urgency health system, and 89% of them work at COVID-19 units. With respect to the level of professional experience, we find that 72% of the sample consists of nurses with less than 5 years of work experience, 16% consists of nurses at their first work experience, and 12% consists of nurses with more than 5 years of experience.

The sample’s description is provided in Table 1.

The narratives highlight a total of 36 events of moral distress, since 6 interviewees report an event that is not representative of a condition of moral distress in accordance with the characteristics described in the literature; thus, it was not possible to proceed with the coding. A total of 29 narratives of events were categorized into the 5 categories of moral events described by literature [18,19,20,21,22,23,24,25,26,27,28,29,30]. A total of seven of the narratives of events analyzed do not satisfy the requisites necessary to fall into the categories of events aforementioned but appear to be representative of a condition of moral distress. Therefore, we created an additional sixth category, specific to the pandemic situation.

The narratives of events are categorized as follows:

***Moral constraint***: A total of 19 narratives of events are attributable to moral constraint events. Nurses remember experiencing situations in which they felt unable to do the right thing, or failed to prevent an action considered morally unjust, due to institutional, contextual, or personal obstacles.

The moral issues narrated for this category of event are:−*Lack of organizational resources*: In their narratives, the nurses remember situations in which external reasons made them feel that it was difficult to provide adequate health care. Some of the reasons were: the lack of beds in COVID-19 units and ICUs; the shortage of health workers and medications; the impossibility of confrontation among healthcare staff members; the inability to move freely and quickly in emergency situations; and the disorganization of hospitals and the lack of time.

*“[...] At that time there were no beds in intensive care, so we continued to do many things that we knew were not enough. The patient had to be intubated, but there was no place to do it. You already knew the patient would get worse but there was nothing you could do to prevent it.”* DNR-RV

−*The poor clinical knowledge on COVID-19*: In their narratives, the nurses remember situations in which they felt unprepared facing the SARS-CoV-2 epidemic; moreover, the virus was manifesting itself differently in each patient, requiring different forms of treatment. This represents an internal and external constraint that confronts nurses with the lack of scientific references for a new and unknown virus.

*“[...] Our lack of knowledge of what COVID-19 was in fact in its clinical manifestations and in its problems, led us to lose many patients and in a very short time and [...], we worked almost like an assembly line.”* NTN-CG

−*Personal protective equipment as an obstacle:* In their narratives, the nurses remember that the obligation to use protective equipment and the obligation of isolation, as well as the deprivation of contact with the outside, represent external constraints which, though indispensable for their safety, were often an impediment to material and emotional assistance to patients.

*“[...] When patients are hospitalized, we are completely harnessed, so they have no external contact, except by telephone with their relatives. Then the state of isolation is added to the difficult situation linked to the protective devices they have to use … and from a psychological point of view they collapse.”* MRI-RV

***Moral tension****:* Two narratives of events are attributable to moral tension events. The nurses remember that, despite knowing what they thought was the right thing to do with the patient, they were unable, for internal reasons, to communicate their position and opinion to the doctors.

The moral issue narrated for this category of event is:−*The inability to have a say*: In their narratives, the nurses remember situations where they thought that more could be done to save some patients. On those occasions, however, they did not express their point of view, as, internally, they perceived that they did not have a space to share their own moral principles, possibly in contrast to those of doctors.

*“A woman with various comorbidities was dying and I was so sorry that she practically died inside a ventilation hood and … without proper” sedation “[...]. There I could, as a health-care professional, insist and ask the doctor to act differently.”* MRS-CG

**Moral conflict**: Two narratives of events are attributable to moral conflict events. Nurses remember situations where what they thought was right to do collided with what someone else thought was right. This generated conflicts over clinical procedures, particularly with doctors. 

The moral issue narrated for this category of event is:−*Non-questionable medical choices:* In the narratives, nurses remember not always being in agreement with the decisions made by doctors. They attempted forms of dialogue, and experienced conflicts in which they were defeated, due to the greater power of decision-making attributed to doctors. This led them to feel that they had not respected the fundamental ethical values of their profession, and to represent the hospital as a sometimes dehumanizing environment.

*“I had a quarrel with the medical staff because of my ethics (which has always been in our ward), which is not to let patients die … or rather, to make them die in the best possible way. When we all understood that the man was dying, suffering and still wearing a ventilation hood, it was useless to continue his suffering, [...] instead, that night the medical staff insisted that he had to keep the ventilation hood on [...]. I asked several times during the night to take it off, but in the end I was unable to win.”* FLR-RV

**Moral dilemma:** Two narratives of events are attributable to moral dilemma events. Nurses remember situations in which dilemmas arose that forced them to choose only one of several possible procedures, all considered equally valid. 

The moral issues narrated for this category of event are:−*The choice of the “right” patient.* In the narratives, the nurses remember that the health emergency led to a large number of infected people in need of hospitalization. This, in turn, led to the need to select those who could actually receive health care. Therefore, the nurses had to choose, from among several patients, the “right” ones who would undergo medical treatment, despite the fact that they all needed it in equal terms.

*“The emergency doctor [...] said to us clearly: “Guys, I have one bed left, so, from the list of 10 people we ought to intubate, we have to pick up only one; then we will see if the others survive the night”. And we were there at the table to figure out who to intubate immediately and who to postpone to the following days instead, provided the patient was still alive. It was chilling enough, yes.”* VRN-RV

−*The choice to take over communication with the family*: In the narratives, nurses remember having to deal with the protocols regarding the issue of healthcare staff–caregivers communication regarding the patient’s clinical conditions. In particular, they recall that they had to choose between waiting for the doctor to take care of communicating with the family, and the instinct to take over this communication to alleviate the concern of family members.

*“[...] Obviously no relatives could come to visit patients, right?! And so we also had to take phone calls. [...] When they call you, you are not in charge of informing the relatives on the patient’s conditions, as the doctor has the responsibility to communicate with them. I wondered how the family members felt; in any case, I had to be honest, but I tried to … reassure them even though I knew that unfortunately the patient was not going to survive. [...]”* FRN-CG

**Moral uncertainty**: Four narratives of events are attributable to moral uncertainty events. Nurses remember situations in which they found it difficult to recognize the right course of action in new and chaotic conditions.

The moral issue narrated for this category of event is:−*The continuous transformation of the virus and of one’s actions.* In the narratives, the nurses remember that the rapid mutation of the virus represented a strong factor of uncertainty about the right actions to take. This uncertainty also originated from the necessary reorganization of the nursing staff in the various departments and hospitals; as a result, nurses were suddenly sent into new departments, where they were asked to reinvent themselves in order to be able to cope with the serious emergency situation. 

*“[...] It is still unclear how to act with COVID. This virus is mutating so fast that drug therapy is good for one patient and less good for another. However, we do our best, we do everything we can. Anyway, I think there is still little clarity about this virus.”* VTL-RV

**Moral compromise**: Seven narratives of events are not attributable to the moral distress categories considered, but allow for the construction of a third category, which we name moral compromise. We describe it as a form of distress that lies between moral constraint and moral uncertainty, without fully falling within the logic of these two categories. In fact, both moral constraint and moral uncertainty predict that there is a right thing to do, whether you know what it is or not. Instead, what we define as “moral compromise” refers to narratives of events in which the nurses not only fail to identify or perform the action deemed right for the patient, but they perceive that there is no morally adequate action in the emergency situation.

The moral issue situation narrated for this category of event is:−*Being unable to do anything to stop the deaths*: In the narratives, nurses remember situations in which they could not do anything to save patients with COVID-19 from death; they gave support, not only material, but above all emotional, to infected patients who suddenly found themselves isolated and without any contact with the outside world.


*“I felt bad, bad, bad. You feel helpless precisely because you can’t … you don’t know what to do because you are helpless, truly helpless [...] there was no time to manage one patient while another died. At the end of the evening I was on the verge of crying, yet I had been in it for a year [...] things could not have gone otherwise, it was inevitable.” CRS-RV*


For each of the six categories of moral events experienced by nurses during the COVID-19 pandemic, we then identified the main emotion, in order to highlight the emotional dimension associated with the narratives of memories of moral issues.

**Moral constraint:** The main emotion associated with the narratives of moral constraint events is *powerlessness.* Nurses feel they have not had the right weapons to fight a war that appears to be lost from the start. Moreover, the lack of external and internal organizational resources related to one’s personal abilities aggravates the sense of powerlessness, linked to the feeling of not being able to do anything to change the course of events determined by the spread of the infection.

**Moral tension:** The main emotion associated with the narratives of moral tension events is *worthlessness,* a sense of inadequacy that nurses experience when they are unable to express their professional opinion about medical choices to make. The atmosphere of great uncertainty, and the confusion of the emergency period, fueled this difficulty in positioning oneself professionally, and taking the floor regarding decisions that seem extremely complex for everyone.

**Moral conflict:** The main emotion associated with the narratives of moral conflict events is *anger.* Nurses experience anger when decisions made by doctors conflict with their own opinion, and are viewed as unfair. Furthermore, such anger is described as a response to doctors who seem to disregard their point of view.

**Moral dilemma**: The main emotion associated with the narratives of moral dilemma events is *sadness*. This experience is connected to the need, inherent in the choices, to let go of the other possible options. In particular, for nurses, the choice can also be between patients to treat, with high risks that the “unchosen” ones will die.

**Moral uncertainty**: The main emotion associated with the narratives of moral uncertainty events is *guilt*. This experience is connected to situations in which nurses did not feel capable of choosing, and felt they did not have the tools to identify and recognize the right course of action in a specific situation.

**Moral compromise**: The main emotion associated with the narratives of moral compromise events is *helplessness.* This experience appears connected to those situations in which nurses feel totally helpless in the face of the anguish of death that also pervades the sphere of morality, making them feel that there is nothing morally right to do.

## 4. Discussion

From a psychological and clinical point of view, Table 2 highlights the way in which the five categories of moral events [18,19,20,21,22,23,24,25,26,27,28,29,30] are defined in the specificity of the narratives of memories of nursing events involved in morally demanding situations of patient care, with attention also paid to the main emotion trajectories connected to the ethical sense-making.

The coding of the nurses’ narratives allows us to identify a sixth category of moral event, specific to the COVID-19 pandemic, that we add to the already existing events in the literature, and which we define as “moral compromise”.

Making use of a focus aimed at reflecting on what are the moral issues of patients’ care concerning the nurses involved during the COVID-19 pandemic, and on what are the main emotional conditions associated with the narratives of their memories, allows us to reflect on some implications that we proposed for the clinical implications of our study.

Narratives of moral constraint events’ memories concern all those impediments to action, mostly deemed just by the nurses, ranging from impediments of a contextual and organizational nature, to impediments related to the absence of knowledge about the virus, and the obligation to wear protective outfits. The process of ethical sense-making, therefore, seems to have been articulated around all those elements that are external to nurses and are seen as obstacles, which prevent nurses from fully exercising their professionalism, resulting in a main emotion of powerlessness.

Narratives of moral tension and moral conflict events’ memories are concerned with the impossibility of having a say in the medical choices and the perception of their unquestionability, respectively. The process of ethical sense-making brings into play issues concerning roles and positions assumed within the organizational hierarchies, which appear particularly rigid in the healthcare sector, especially in times of emergency. On the one hand, the aspects of tension refer to the emotion of uselessness and, therefore, to the internal feeling of not recognizing legitimacy in expressing opinions on significant medical choices. On the other hand, the aspects of conflict refer to the emotion of anger towards doctors who, though seeming to offer space for discussion, do not consider integrating the opinions of nurses when making decisions. There seems to be almost a continuum between tension and moral conflict, ranging from the impossibility that nurses feel to position themselves in the decision-making process, to the futility of doing so. We point out that the emotion of uselessness does not appear in the literature as a typical emotion of moral tension events; instead, it is characterized by anger, frustration, and helplessness [18]. In the pandemic context, the level of self-worth and self-esteem appears undermined when nurses feel they are unable to involve themselves in a moral conflict, probably due to the overload of media and social expectations regarding their role.

Narratives of moral dilemma events’ memories concern the difficulty of selecting the few patients to treat, and the choice of the staff entitled to communicate with family members. The sense-making process is articulated around the losses involved in the need to choose, where choice is configured as a primary condition of professional action. The main emotion of sadness makes us think that nurses working during the pandemic had to give up aspects they consider significant in their job. Nurses had to accept that they cannot always express the moral values related to the humanization of care, the respect for the patient as an individual, and not just as a sick person, and the emotional involvement in the relationship with caregivers. This constitutes a significant loss for them, which is the object of elaboration and reflection through the narration of their memories.

Narratives of morally uncertain events’ memories are concerned with the difficulty of recognizing the right action to take in ever-changing and chaotic situations. It looks as if the sense-making process moves in an undifferentiated way among never definable choices, due to the absence of interpretative coordinates of the phenomenon. The main emotion of guilt, which nurses address in themselves and not in elements of the context, appears to be very significant. It seems that in an emergency situation, in which every scientific, organizational, and working practice reference is put into crisis, a game between impotence and omnipotence emerges in salience, which inevitably leads to responsibility and guilt for not finding solutions and “not knowing what to do”.

Narratives of moral compromise events’ memories concern the impossibility of action in stopping the constant deaths of patients. The ethical sense-making process is articulated around the subject “Pandemic”, and the feeling that it is an unethical event in itself. The main emotion of helplessness seems to be linked to a crisis of one’s professional and moral identity, and to a profound compromise of one’s “moral agency” [54], all flattened to a condition of helplessness in the face of death.

The impact with a potentially traumatic situation, in which nurses are required to act with no psychological and material resources to think about what was going on, probably contributed to emotions of deep distress, despondency, and annihilation related to the certainty that there was nothing that could be done.

We imagine that the period in which the stories were collected, that is, the initial phase of the health emergency, instilled memories of moral compromise in nurses.

## 5. Conclusions

This study allows us to identify the process of ethical sense-making concerning psychological, relational, and contextual dimensions, indicating that nurses’ moral distress concerns both “internal” and “external” aspects related to their professional context, and their relationships with colleagues.

Despite the limitations of the study, mainly due to the fact that the sample was not equally distributed in the different categories of events, making it difficult to generalize the results, these reflections suggest, from a clinical intervention perspective, some trajectories. In particular, they underline the need to work on multiple axes to deal with moral distress, and promote well-being and moral resilience among nurses [55,56] in order to ensure quality healthcare, even in emergency situations.

From a clinical point of view, it is important to implement organizational health resources, promote dialogue between care team members in which nurses have the opportunity to express their views, [20] and support nurses in self-care practices in order to prevent burn-out and turn-over phenomena, through the presence of psychological support interventions [57,58,59,60].

## Figures and Tables

**Table 1 ijerph-19-08349-t001:** Sociodemographic characteristics.

**Sex**	**%**
Male	38%
Female	62%
**Years of Experience**	**%**
0	16%
<5	72%
>5	12%
**Geographical Location**	**%**
North (Italy)	51%
Center (Italy)	5%
South (Italy)	44%
**Work Setting**	**%**
COVID-19 Unit	11%
Emergency–urgency health system	89%

**Table 2 ijerph-19-08349-t002:** Moral events, COVID-19 moral issues, main emotions.

Moral Events	COVID-19 Moral Issues	Main Emotions
*Moral constraint*	Lack of organizational resources Poor clinical knowledge of COVID-19 Personal protective equipment as obstacle	*Powerlessness*: feeling powerless and deprived of organizational and internal resources in the face of the war against the spreading virus
*Moral tension*	The inability to have a say	*Worthlessness:* feeling inadequate in expressing their opinion on medical choices to be made in an emergency
*Moral conflict*	Non-questionable medical choices	*Anger:* feeling angry in response to medical decisions that conflict with their own opinion
*Moral dilemma*	The choice of the “right” patient The choice to take over communication with the family	*Sadness*: feeling sad because choosing implies the loss of a possible option, or even a patient
*Moral uncertainty*	The continuous transformation of the virus and of one’s actions	*Guilty*: feeling guilty of not being able to make any decisions in situations of great confusion
*Moral compromise*	Being unable to do anything to stop the deaths	*Helplessness:* feeling helpless in the face of a situation deemed immoral tout court

## Data Availability

Not applicable.

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
