# Peer review of "Moral Distress Events and Emotional Trajectories in Nursing Narratives during the COVID-19 Pandemic"

_ijerph, 2022, doi:10.3390/ijerph19148349_

Round 1

Reviewer 1 Report

This is an important study and should be published to help professionals and the public understand the impact on health care workers under the impossible constraints of Covid-19. 

I would recommend explaining the references to "a helmet" in two of the quoted interviews as that reference is not familiar to me in the health care context. 

For further research using these interviews, I would recommend the readers explore the concept of "moral injury," which describes perceived impact to the self as a moral agent under situations of constraint similar to those described in this research. This would connect the researchers with a significant pre-existing literature and a term that is becoming familiar even to non-specialists (for example, it was recently used in the New York Times.) 

See: https://www.thelancet.com/journals/lanpsy/article/PIIS2215-0366(21)00113-9/fulltext

Author Response

Thanks for your valuable suggestions and directions.
I am glad that you consider the article useful for the scientific world.
I have clarified and changed the word "helmet" with "ventilation hood" in the
text. It refers to the venitilatory therapy of subjects in intensive care.
Very interesting is the construct of the moral injury.
I will deepen it for future research. Thanks

Reviewer 2 Report

I would like to congratulate the team of authors for the study carried out on a topic that is becoming more and more necessary, such as making visible and showing the mental health and psychological problems generated by the Covid-19 pandemic in nurses.

After reviewing the article, I note my suggestions and recommendations:

§  The abstract should conform to a maximum of 200 words.

§  It is recommended to include in the Material and Methods section, the design of the study and to expand the section a little more, explaining the script of the ad-hoc narrative interview.

§  In the results it is recommended to insert explanatory figures that provide visual information.

§  It is recommended that the wording of lines 427 to 433 be modified, as they are confused with the final conclusion of the study.

§  It is recommended that a synthesis of the conclusion be made, as it is very extensive.

§  The bibliographic references should be revised, some are more than 10 years old.

Kind regards.

Author Response

Thank you for appreciating my manuscript and for the clear revision directions.
- the abstract is 195 words long
- in the materials and methods section we have added the interview areas with narrative prompts
- the results have been summarized in a table that you can find in the "supplementary materials" uploaded to the submission
- we reworked discussions and conclusions, summarizing the conclusions section
- we have reviewed the bibliographical references, putting the most recent ones where possible. Some "classic" authors of reference for narrative methodology are important for us to cite as a theoretical framework